# Melatonin Treatment in Kidney Diseases

**DOI:** 10.3390/cells12060838

**Published:** 2023-03-08

**Authors:** Magdalena Markowska, Stanisław Niemczyk, Katarzyna Romejko

**Affiliations:** Department of Internal Diseases, Nephrology and Dialysis, Military Institute of Medicine, 128 Szaserów Street, 04-141 Warsaw, Poland

**Keywords:** melatonin, kidney diseases, renoprotective effect, chronic kidney disease, nephrotoxicity, acute kidney injury

## Abstract

Melatonin is a neurohormone that is mainly secreted by the pineal gland. It coordinates the work of the superior biological clock and consequently affects many processes in the human body. Disorders of the waking and sleeping period result in nervous system imbalance and generate metabolic and endocrine derangements. The purpose of this review is to provide information regarding the potential benefits of melatonin use, particularly in kidney diseases. The impact on the cardiovascular system, diabetes, and homeostasis causes melatonin to be indirectly connected to kidney function and quality of life in people with chronic kidney disease. Moreover, there are numerous reports showing that melatonin plays a role as an antioxidant, free radical scavenger, and cytoprotective agent. This means that the supplementation of melatonin can be helpful in almost every type of kidney injury because inflammation, apoptosis, and oxidative stress occur, regardless of the mechanism. The administration of melatonin has a renoprotective effect and inhibits the progression of complications connected to renal failure. It is very important that exogenous melatonin supplementation is well tolerated and that the number of side effects caused by this type of treatment is low.

## 1. Introduction

Melatonin, discovered in the late 1950s, is a pleiotropic neurohormone that is mainly produced in the pineal gland. While it is released from different tissues, it acts also as a local regulatory molecule [1]. The elementary role of melatonin is to transmit information concerning the daily cycle of light and darkness to the different parts of the human body, which ultimately affects the functioning of the entire organism [2]. However, there are many reports showing that this is not the only mechanism and function of this particle. It has been proven that melatonin also takes part in antioxidative, anti-inflammatory, antiapoptotic, and immune processes [1,3,4,5,6,7,8,9,10,11,12,13]. Moreover, melatonin participates in the detoxification of free radicals, bone formation, reproduction, and body mass regulation and has an influence on cardiovascular homeostasis [14,15,16,17]. The renoprotective effect of melatonin has been the subject of reports in the last decade that have found that melatonin not only ameliorates sleep disorders in patients with chronic kidney disease (CKD) but also has a beneficial effect on blood pressure and provides protection in oxidative stress and inflammation [18,19,20], which occur in a wide variety of kidney injuries such as CKD, glomerulonephritis, contrast-induced kidney injury, drug-induced nephrotoxicity, and acute ischemia–reperfusion injury. This review summarizes the physiology of action and the final effects of melatonin treatment in different types of kidney injuries.

## 2. The Biosynthesis and Metabolism of Melatonin

Melatonin is a neurohormone whose main source is the pineal gland [21]. The production of melatonin is dependent on the light/dark cycle. Interestingly, light can either suppress or initiate melatonin synthesis [22,23]. When it is received by the retina, the signal is sent to the suprachiasmatic nuclei of the hypothalamus [24]. Later, it goes through the upper part of the cervical spinal cord, where the synapses, which are connections with preganglionic cell bodies of the superior cervical ganglia, are located. Finally, it reaches the pineal gland. The signal is mainly transmitted by a neuromediator called norepinephrine [22]. It binds with beta-adrenergic receptors and stimulates the pineal gland to produce melatonin [24]. The synthesis starts with the hydroxylation of the amino acid tryptophan (by tryptophan-5-hydroxylase) to 5-hydroxytryptophan, then decarboxylation (by 5-hydroxytryptophan decarboxylase) to serotonin. This is followed by the acetylation of serotonin to form *N*-acetylserotonin (by arylalkylamine-*N*-acetyltransferase). The concluding stage is the methylation of *N*-acetylserotonin by acetylserotonin-*O*-methyltransferase. After all above-mentioned reactions, it is possible to obtain the final product, which is melatonin [17] (Figure 1).

The production process of melatonin takes place mostly during the night [22,23]. Both the synthesis and secretion length are directly dependent on the duration of the sleep period. It is a time-based transmitter that conveys information about the round-the-clock cycle of light and darkness to the body [25,26]. However, it is important to emphasize that the pineal gland is not the only place where melatonin is synthesized. It is also produced by retinal photoreceptors [27], the gastrointestinal tract [28,29], bone marrow [30], the liver, the kidneys, the thyroid, the pancreas, the thymus, the spleen, the carotid body, the reproductive tract, and endothelial cells. Human skin is also a place where all enzymes involved in production process are expressed [10]. There are two G-protein-coupled melatonin receptors: MT_1_ and MT_2_ [31]. After their activation, the intracellular level of the second messenger cyclic adenosine monophosphate (cAMP) is decreased. The result is a modification of signaling pathways below protein kinases A and C, and a cAMP reaction with an element-binding protein [10,32]. Melatonin receptors are widespread. Most of them are distributed in the central nervous system, but they are also located in peripheral body parts such as the retina, cerebral and peripheral arteries, the kidneys, the pancreas, the adrenal cortex, the testes, and immune cells [33,34].

## 3. The Biological Role of Melatonin

### 3.1. The Nervous System

It is well known that the concentration changes of melatonin take part in sleep–wake cycle disorders, mood disturbances, disabilities of cognitive skills, troubles with learning and memory problems, protection of the nervous system, drug abuse, and cancer processes. Therapies based on pharmacological agonists of melatonin (agomelatine, ramelteon, and tasimelteon), which also affect MT1/MT2 receptors, have been the subject of research interest in recent years [35]. Melatonin can be a potential course of action for novel antidepressants, which affect the concentrations of neurotrophins or neurotransmitters. In addition, they cause a reduction in the proinflammatory cytokine level in the serum [36]. The neuroprotective effect of melatonin is used in the treatment of Alzheimer’s, Parkinson’s, and Huntington’s diseases as well as amyotrophic lateral sclerosis, stroke, and brain trauma [7,37]. Due to its antioxidant properties, melatonin acts as a scavenger of free radicals and regulates numerous reactions at the molecular level, including oxidative stress, inflammation, and apoptosis [38,39]. It has also been documented that melatonin is an inhibitor of calpain, whose activity is significant in the pathogenesis of many central nervous system disorders [40].

### 3.2. The Immune System

Another important role of melatonin is its ability to immunomodulate and strengthen immune surveillance [41]. It stimulates the production of different lines of cells involved both in humoral and cell-mediated immunity, such as macrophages, natural killer cells, and CD4+ cells, and affects the synthesis of a wide variety of cytokines [42]. The direct anti-viral and anti-bacterial effects of melatonin have been documented [43,44,45]. During severe infections, the administration of melatonin has been found to have immunomodulatory, antioxidative, and cytoprotective functions [44,46,47]. It has been proven that due to its beneficial pleiotropic effects, the administration of melatonin reduces mortality in both viral and bacterial inflammation [48]. Considering the evidence supporting the role of this hormone in directing oxidative stress and inflammatory processes, as well as the management of immune reactions, examinations involving patients with viral infections caused by Severe Acute Respiratory Syndrome Coronavirus 2 (SARS-CoV-2) have also been conducted. They showed that the administration of melatonin as an adjuvant therapy might be beneficial and that it should be considered during Coronavirus Disease 2019 (COVID-19) [6,49,50].

### 3.3. The Gastrointestinal Tract

Melatonin in the digestive system, besides its antioxidant effect and ability to stimulate the immune system, reduces the secretion of hydrochloric acid, enhances the regeneration of the epithelium, and increases microcirculation. All of these functions make melatonin one of the therapeutic options for preventing different diseases of gastrointestinal tract, e.g., colorectal cancer, ulcerative colitis, gastric ulcers, and irritable bowel syndrome. It may also be helpful during treatment [51,52]. It has been documented that melatonin supplementation results in the complete remission of gastroesophageal reflux disease. It has a protective role against acute and chronic irritants that affect the esophagus and stomach. It is also effective in healing ulcers [53]. Moreover, some studies have confirmed that melatonin has strong supporting effects on hepatocytes in the prevention of non-alcoholic steatohepatitis (NASH) [54].

### 3.4. The Respiratory System

A positive therapeutic potential of melatonin has been also discovered in patients suffering from pulmonary disorders. Melatonin prevents inflammation, eliminates several oxygen-derived reactants, detoxifies nitric oxide, and takes part in apoptosis, including in cancer cells. It also inhibits the proliferation of these malignant cells [55,56,57]. In a fairly similar mechanism, melatonin influences the restriction of pulmonary fibrosis. It reduces endothelial cell proliferation, invasion, and migration [55,58]. Moreover, it can minimize the gathering of inflammatory cells and reduce the expression of inflammatory mediators, such as cyclooxygenase-2. The amount of proinflammatory cytokines also decreases, which consequently leads to the inhibition of cellular proliferation [59]. The role of melatonin during respiratory tract infections is also important and is discussed in the section regarding the immune system.

### 3.5. Endocrinology and Gynecology

Clinical trials with melatonin administration have been conducted on animal and human groups. According to contemporary knowledge, melatonin can improve fertility, oocyte quality, maturation, and the number of embryos [14,60]. Moreover, a positive effect during pregnancy has been suggested. The protection of neurogenesis, a supportive impact on the placenta, and a reduction in oxidative stress are mechanisms that increase the reproductive rate and improve embryo–fetal development [60,61]. Reactive oxygen species cause disturbances during pregnancy. They are also responsible for complications in the perinatal period. Melatonin is a scavenger of free radicals. It has also antioxidative and cytoprotective abilities. There is a possibility that it can be crucial for a successful pregnancy. Not only is the role of melatonin in human gendering important, its support is also necessary when neonatal pathologies occur [62,63]. Melatonin is a supervisor during the process of deoxyribonucleic acid (DNA) methylation and histone alteration. In this way, radical changes in gene expression are avoided. The fetus is protected from the occurrence of pathologies. An insufficient concentration of melatonin during pregnancy can leave endocrine disorders in the genetic code during early ontogenesis, which subsequently develop in childhood [64].

### 3.6. Other Functions

Some additional, very important functions of melatonin in the human body are described in the section about kidney diseases below. The roles of this hormone in different parts of human body are summarized in Figure 2.

## 4. The Role of Melatonin in Kidney Diseases

### 4.1. The Role of Melatonin in Chronic Kidney Disease

Chronic kidney disease (CKD) is currently one of the leading public health problems worldwide. It affects almost 13% of the world population [65]. CKD is associated with many complications, such as malnutrition; anemia; hyperlipidemia; overhydration; and endocrine, mineral bone, and metabolic disorders [65,66,67]. Patients with CKD, especially those who are treated with renal replacement therapy, frequently suffer from sleep disturbances [68]. It has been reported that 80% of end-stage renal disease patients complain about sleep disorders [69,70,71,72]. The complications of CKD intensify insomnia as well as depression, anxiety, and itch, which often affect patients with decreased kidney function [73,74,75]. A chronic deficit of sleep can lead to metabolic and endocrine disorders, e.g., diabetes, obesity, or hypertension [76,77,78,79,80,81]. Behavioral interventions and pharmacological treatments are often not sufficient [82]. There is convincing evidence that melatonin efficiently accelerates falling asleep; regulates the duration of wake times; and improves concentration, reflexes, and cognitive functions [83,84,85,86,87,88]. It has been documented that hemodialysis patients suffer from disturbances in the diurnal rhythmicity of the sleep–wake cycle and melatonin concentrations [89,90]. Melatonin synchronizes the circadian rhythms, improves the quality of sleep, and is involved in neuronal survival [91]. It has been found that it prevents further implications of sleeplessness such as neurodegenerative diseases [83,92]. Melatonin administration is recommended for different types of sleep disorders, as it synchronizes the circadian rhythms, depending on the time of day when the drug is taken [93]. Edalat-Nejad et al. carried out a 6-week randomized, double-blind, cross-over clinical trial in hemodialysis patients. Melatonin was administered to patients at bedtime. As a result, the quality of sleep improved [94]. It should not be overlooked that melatonin supplementation is well tolerated, with a small number of side effects [87,88,91]. In the available clinical trials, exogenous melatonin is an effective drug with a low risk of dangerous side effects in patients with CKD [68,95,96]. Ramelteon, a melatonin-receptor agonist, is also approved for the treatment of insomnia. It has been reported that it is safe and effective [97].

The dysregulation of the circadian rhythm is connected with a higher risk of cardiovascular events [98,99,100,101,102]. The strong association between cardiac reactions and time of day is the reason why myocardial infarction (MI), sudden cardiac death, and ischemic stroke are more likely to occur in the early morning [103,104,105]. Moreover, there have also been examinations assessing healing after MI, depending on the circadian rhythm [106,107]. It was proven that any disruptions result in alterations in immune responses, which are crucial for scar formation and the functioning of the heart in the future. During a clinical trial with mice in the proliferative phase, 1 week after MI, less blood vessels formed in the infarcted area compared with the control group. Echocardiography after 14 days showed increased left ventricular dilation and infarct expansion [106]. This proves that the stabilization of the biological clock is needed to maintain homeostasis in the whole organism and support recovery [108]. It has been found that melatonin provides protection by activating silent information regulator 1 (SIRT1), which acts in a receptor-dependent manner. It causes a reduction in apoptotic protein expression and an increase in antiapoptotic protein [109]. Moreover, melatonin protects cells from reactive oxygen species [110,111]. It results in an improvement in cardiac function, a reduction in oxidative damage, and a decline in myocardial apoptosis [112,113]. Oxidative stress causes cellular injuries in the vascular system and induces inflammatory processes [114,115]. It is directly involved in the pathogenesis of cardiovascular diseases [98]. Among patients in the early stages of chronic kidney disease, the incidence of cardiovascular events is significantly higher. Moreover, the prevalence rate increases commensurably with the advancement of kidney function deterioration [116]. Unfortunately, cardiovascular diseases are a frequent reason for the increased morbidity and mortality in this group of patients [43,117,118,119,120]. New therapies are sought to prevent cardiovascular complications in CKD patients. Antioxidant therapies and clinical trials are being conducted [121]. Melatonin has been documented to participate in controlling oxidative stress and has been shown to have a positive influence on the cardiovascular system [98,122,123].

Atherosclerosis and hypertension are also frequent complications of CKD [120,124]. On the other hand, hypertension can lead to a deterioration of kidney function and is the second leading cause of end-stage renal disease [125]. There are several mechanisms of hypertensive renal damage, such as the renin–angiotensin–aldosterone system (RAAS), oxidative stress, endothelial dysfunction, and genetic determinants. Inflammation and fibrosis lead to glomerular sclerosis, tubular atrophy, and interstitial fibrosis [126]. In addition, advanced kidney disease can cause difficulties in blood pressure normalization [127]. It has been proven that melatonin may play a role in reducing blood pressure during the day and night by influencing changes in the endothelium and in the functioning of the autonomic nervous system and the renin–angiotensin system. Moreover, it reduces oxidative stress [68,128]. A clinical trial whose aim was to observe the regulation of blood pressure by melatonin showed that after pinealectomy and in other cases when the concentration of melatonin in the plasma decreased, patients should receive melatonin supplementation in order to maintain the correct blood pressure values [129]. Melatonin also takes part in delaying atherosclerosis [130,131,132]. The degree of changes in blood vessels is higher in proportion to the progression of renal disease. The pathogenesis is connected to systemic inflammation and the increased amount of reactive oxygen species [120,133]. Increased carotid artery intima–media thickness, carotid arterial wall stiffness, and coronary artery calcification are also common in children with chronic kidney disease [134]. The study by Zhang proved that melatonin reduces the atherosclerotic plaque in the aorta [135]. This is very important because the stage of atherosclerosis is a strong predictor of the mortality rate due to cardiovascular disease in patients with CKD [133,136]. Anti-inflammatory treatment strategies could be beneficial [137].

Obesity and diabetes are states that often coexist with CKD and deepen kidney injury [138,139,140]. Diabetic nephropathy occurs in one third of diabetic patients [141]. Adipocytes trigger the release of proinflammatory cytokines [138,142]. Serum concentrations of C-reactive protein, adiponectin, resist in, interleukin-6, tumor necrosis factor-alpha, monocyte chemoattractant protein-1, and CD68 are responsible for chronic inflammation [143]. While the levels of these molecules are high in blood circulation, they bind with receptors that are located in the cell membranes of renal tissues. As a result of this connection, the kidneys are damaged [144]. Melatonin, with its antioxidant and anti-inflammatory abilities, influences this process through several mechanisms such as NF-κB amelioration and NLRP3-inflammasome signals, reducing proinflammatory protein expression in the serum, regulating metabolic conversion and the energy balance, activating receptors in adipocytes, and sensitizing adipocytes to insulin and leptin [145,146]. Obesity is associated with hemodynamic, structural, and histological renal changes [138]. The hypertrophy of glomeruli and tubules is observed. Focal segmental glomerulosclerosis, or bulbous sclerosis, is also possible. The gradual progression of nephropathy is connected with renal hemodynamic changes, insulin resistance, and lipid metabolism disorders [147]. These effects are crucial for the progression of kidney diseases and are also associated with the advancement of cardiovascular complications. Melatonin has a hypolipidemic impact by enhancing endogenous cholesterol clearance mechanisms [148,149]. It takes part in bilirubin acid production and suppresses low-density lipoprotein receptor activity. It also causes an increase in the level of irisin and accelerates cholesterol excretion in the feces [150]. In an experiment with mice, lipid accumulation varied considerably during the administration of melatonin. It was reduced, and the expression of lipid metabolic genes was minimized [151]. Melatonin not only influences the causes of hypercholesteremia but also protects tissues from the toxic effects of oxidized lipoproteins [152]. It is assumed that this is an effect of melatonin’s impact on cell membranes [153,154]. Melatonin treatment has also been proven to be beneficial in diabetic patients [155,156,157]. Alack of this hormone causes a reduction in glucose transporter type 4 (GLUT4) gene expression, which results in the development of glucose intolerance and insulin resistance [156]. Moreover, melatonin enhances insulin secretion and β-cell existence. Islet sensitivity to cAMP is higher during melatonin supplementation, and it results in an increase in insulin secretion [158]. Simultaneously, melatonin stimulates glucagon release [155]. It has been documented that there are some occasional changes in the coding of the melatonin receptor gene MTNR1B [155,157]. This consequently causes an inhibition of melatonin binding and information transmission. This is ultimately associated with a higher risk of type 2 diabetes mellitus [155,156,157,159,160,161]. A reduced level of melatonin is also implicated in the pathogenesis of type 2 diabetes [160,161]. A study by Mok that included numerous clinical trials with the use of melatonin supplementation in patients with diabetes proved that melatonin administration may be a new therapeutic method to improve the diabetic condition and reduce the prevalence of diabetic complications [159,161]. 

### 4.2. The Role of Melatonin in Glomerulonephritis

Glomerulonephritis is a collection of different types of kidney damage at the level of the glomerulus. This group has a variety of causative factors. However, most of them are the consequences of immune processes [162].

The etiopathogenesis of lupus nephritis is not well understood. It is connected to the activation of the NLRP3 inflammasome, which is a member of the NLR family (NOD-like receptors). It is involved in the synthesis of proinflammatory cytokines [163,164]. Histological examination reveals severe renal damage. The widening of tubules and capillaries and the extendedness of the mesangial matrix can be observed. It is manifested by hypertrophy of the mesangium, glomerular atrophy, and the thickening of the capillary walls and basement membrane. It has been proven that the described alterations are reduced during melatonin treatment [165]. Melatonin, with its important antifibrotic, antioxidative, anti-inflammatory, and pro- and anti-apoptotic effects, inhibits lupus-related nephropathy and gives a chance to avoid the complications of autoimmune diseases [166,167].

Focal segmental glomerulosclerosis (FSGS), which is another type of glomerulonephritis, depends on focal and segmental glomerulosclerosis with tubular involution and interstitial fibrosis [168]. One of the discovered candidate genes, which allowed researchers to know the exact mechanism of FSGS and find guidelines for the diagnosis and therapy of the discussed disease, is Melatonin Receptor 1A (MTNR1A) [169].

The efficacy of melatonin therapy was also assessed in a group of patients with membranous nephropathy. There were several beneficial aspects during melatonin treatment: a significant reduction in proteinuria, an improvement of glomerular damage, a decreased deposition of immunocomplex, a decrease in the subpopulation of CD19+ B cells, and proinflammatory cytokines with a one-step increase in the expression of anti-inflammatory cytokines. Moreover, the secretion of reactive oxygen species was minimized. All these findings show that melatonin treatment prevents the development of membranous nephropathy using many different pathways [170].

### 4.3. The Role of Melatonin in Contrast-Induced Kidney Injury

Contrast-induced acute kidney injury (CI-AKI) is a condition in which a progressive deterioration of kidney function is observed a few days after contrast administration. Precisely, it should be described as an increase in serum creatinine ≥0.3 mg/dL or ≥1.5–1.9 times above normal in the 48–72h following contrast medium administration [171]. This is consistent with the definition of AKI in *The Kidney Disease: Improving Global Outcomes*. This situation is a result of the direct impact of contrast medium on the kidneys. Nephrotoxicity is manifested by damage to tubular epithelial cells. Moreover, vasoactive molecules are released. They stimulate oxidative stress, leading to ischemic injury [172]. The direct cytotoxic effect and the hemodynamic alterations are two key mechanisms in the pathophysiology of CI-AKI [173]. Endothelial cell apoptosis and inflammation also occur during CI-AKI [174]. All these changes lead to eGFR reduction [175]. Multiple pharmacologic strategies, usually in connection with maintaining proper hydration, are still used to prevent CI-AKI [176]. In many trials, various techniques to avoid injuring renal cells with free radicals were attempted, but the roles of most of them, including antioxidant agents, remain unclear [177,178]. The most extensively studied techniques are N-acetylcysteine, which removes reactive forms of oxygen from the organism, and nitric oxide, which dilates the vessels [178]. The role of free radicals is crucial in renal vasoconstriction. Numerous data have shown the renoprotective effect of melatonin, as assessed by the normalization of creatinine and urea in the serum, positive alterations in histological examination of renal tissues, and decreases in the levels of early indicators of kidney injury and neutrophil-gelatinase-associated lipid (NGAL) injury [179]. The use of melatonin as a premedication in examinations with contrast caused a relevant enhancement of the expression of Sirt3 and decreased the ac-SOD2 K68 level. As a result, oxidative stress was significantly decreased [180]. Taking into account the anti-inflammatory function of melatonin, it may play a role as one of the effective preventive strategies for CI-AKI [180].

### 4.4. The Role of Melatonin in Treatment-Induced Nephrotoxicity

Acute kidney injury is a common complication of drug administration. Drug nephrotoxicity is divided into two types, depending on the pathomechanism. The first one is mediated by inflammation and is commonly referred to as acute interstitial nephritis. It is usually caused by an allergic reaction. The second type is known as toxic acute tubular necrosis. It occurs when the pharmacologic agents or their metabolites act as direct tubular toxins [181]. There are numerous factors that influence the kidney response to pharmacological treatment. Some of them depend on patient (gender, age, and genes), drug (dose, solubility, and direct nephrotoxic effect) and kidney factors (blood flow and proximal tubular uptake of the drug) [182]. The most common risk factors among patients suffering from drug-induced nephropathy include an elderly age, dehydration, pre-existing renal dysfunction, and the simultaneous use of other nephrotoxins [183]. Unfortunately, incomplete renal recovery is observed in one third of patients with drug-induced nephrotoxicity. The duration of injury prior to diagnosis is important [184]. It is not easy to observe the early symptoms of kidney damage because minor changes in renal function are often clinically asymptomatic [185]. The metabolism of drugs is a process coordinated by multiple renal enzyme systems, including CYP450 and flavin-containing monooxygenases. During biotransformation, toxic metabolites and reactive oxygen species are produced. The accumulation of these molecules leads to oxidative stress. All these reactions contribute to kidney injury [186]. There are several well-documented drugs that cause acute kidney disease. The prolonged administration of cyclosporine A (CsA) and alterations in the structures of kidneys were studied using electron microscopy and morphometry. Apoptosis and the necrosis of proximal tubules with dislocated brush borders, swollen mitochondria, multiple lysosomes, unformed basement membranes of the glomeruli, and atypical mesangial matrices were observed. Treatment with melatonin partially prevented these disturbances/disorders. Melatonin also attenuated damage caused by CsA [187]. Renal fibrosis was observed following treatment with CsA and exposure to carbon tetrachloride (CCl4). Melatonin minimized the accumulation of leukocytes by reducing the expression of iNOS and p38-mitogen-activated protein kinase (MAPK). It also protected kidneys against the flow of mononuclear cells and fibrosis, which are induced by CCl4 [188]. Nephrotoxicity is also the main adverse outcome of vancomycin administration. There is also a connection between vancomycin and oxidative stress. After the administration of vancomycin, the production of intracellular reactive oxygen forms in LLC-PK1 cells located in the renal tubules was higher and caused cellular apoptosis [189]. The supplementation of melatonin reduced the episodes of acute kidney injury during treatment with this antibiotic [190]. This is similar to cisplatin therapy. The number of renal complications is relatively high [191]. The apoptosis and necroptosis of renal cells are the pathomechanisms of kidney injury during treatment with cisplatin. The anti-inflammatory properties of melatonin allow it to inhibit these processes. This is possible due to the upregulation of RIPK1, and the RIPK3-multiprotein complex, which plays a crucial regulatory function in the initiation of cell death, is significantly attenuated by melatonin [192].

When it comes to different types of therapy, drugs are not the only treatment that can cause nephrotoxicity. Radiation is also limited by the possibility of kidney injury. It depends on the dose and time of exposure. It is estimated that after 6 months of radiotherapy, patients develop latent acute nephritis, and chronic nephritis may occur after 18 months of the treatment. Melatonin scavenges hydroxyl radicals, inhibits nitric oxide synthase, and increases the stimulation of antioxidant enzymes with significant functions in the organism, including superoxide dismutase and glutathione dismutase. The protective effect of this hormone was assessed using both light and electron microscopy [193].

### 4.5. The Role of Melatonin in Acute Ischemia–Reperfusion Injury

Surgical procedures also sometimes result in acute kidney injury. Cardiac surgery and renal transplantation are operations during which melatonin treatment has a positive effect on kidney deterioration. Cardiac surgery causes renal dysfunction in approximately 7.7% of patients [194]. Acute kidney injury during cardiac surgery is associated with higher morbidity and mortality and often prolongs hospitalization. There are several factors that lead to renal dysfunction after cardiac surgery, such as nonpulsatile blood flow; catecholamines and mediators of inflammation that circulate in the blood; and kidney injury caused by an embolus and free hemoglobin, which is released from destroyed erythrocytes. All of them lead to renal complications [195]. The time of ischemia correlates with the degree of kidney damage and its irreversibility (Table 1) [196].

There are several examinations that suggest a potential therapeutic effect of melatonin in minimizing renal injury during ischemia and reperfusion [197,198,199]. A histopathological assessment of kidney injury after melatonin treatment showed that the damage was smaller [197]. This may be explained by multiple pathways, e.g., a lower level of the lipid peroxidation marker malondialdehyde, higher activity of superoxide dismutase and catalase, reduced apoptosis due to minimized DNA damage, and a suppression of inflammation, which is expressed by reductions in the concentrations of tumor necrosis factor-alpha, interleukin-1β, nuclear factor kappa B, kidney injury molecule-1, *IL-18*, matrix metalloproteinase, and neutrophil-gelatinase-associated lipocalin [200].

The anti-inflammatory and anti-oxidative functions of melatonin have also been observed among kidney transplant recipients. The administration of melatonin led to improved kidney function after renal transplantation. It was manifested by serum levels of biomarkers such as neutrophil-gelatinase-associated lipocalin, whose concentration was significantly decreased after the administration of melatonin [201]. In addition, the inhibition of oxidative stress, apoptosis, and the secretion of proinflammatory cytokines and the impedance of neutrophil and macrophage accumulation as well as increased autophagic outflow were observed during melatonin treatment [197]. In conclusion, melatonin, with its antioxidant effects, is potent for inverting the consequences of acute kidney ischemia [202].

## 5. Summary

Melatonin has both direct and indirect influences on the well-being of patients with CKD. Firstly, it regulates the day and night cycle and maintains the proper quality of sleep. Sleeping disorders lead to depression and behavioral complications. Deficient sleep is connected to physical weakness, an increased level of aggression, and attention disorders. It also restricts the social life and personal development due to memory deterioration or reductions in performance on physical examinations. Furthermore, abnormalities in melatonin production or secretion are connected to pathologies of the nervous system, such as Alzheimer’s and Parkinson’s diseases. Besides controlling the circadian rhythm, the melatonin profile in human physiology has certain additional effects related to the glucose balance, the control of blood pressure, phosphocalcic metabolism, and hemostasis. The process of kidney degeneration is often an implication of the frequent prevalence of arterial hypertension, diabetes mellitus, atherosclerosis, and obesity. Melatonin has been proven to have beneficial effects in all of these complications. Melatonin also plays a direct renoprotective role. Moreover, it can be helpful in almost every type of kidney injury because inflammation, apoptosis, and oxidative stress occur without regard to the mechanism. Melatonin regulates mitochondrial metabolism and ATP production and protects mitochondria. It inactivates free radicals by attaching one or more electrons and thus reduces oxidative stress. Due to these mechanisms, melatonin enables normal mitochondrial functions and protects patients from subsequent apoptotic implications and the death of kidney cells. The role of this hormone in kidney disease is summarized in Figure 3.

So far, there have been many studies of exogenous melatonin administration to animals. However, the number of human studies with the use of melatonin is not high, but it is increasing. According to the available clinic trials, melatonin can improve the quality of life and prolong survival in patients with CKD.

## Figures and Tables

**Figure 1 cells-12-00838-f001:**
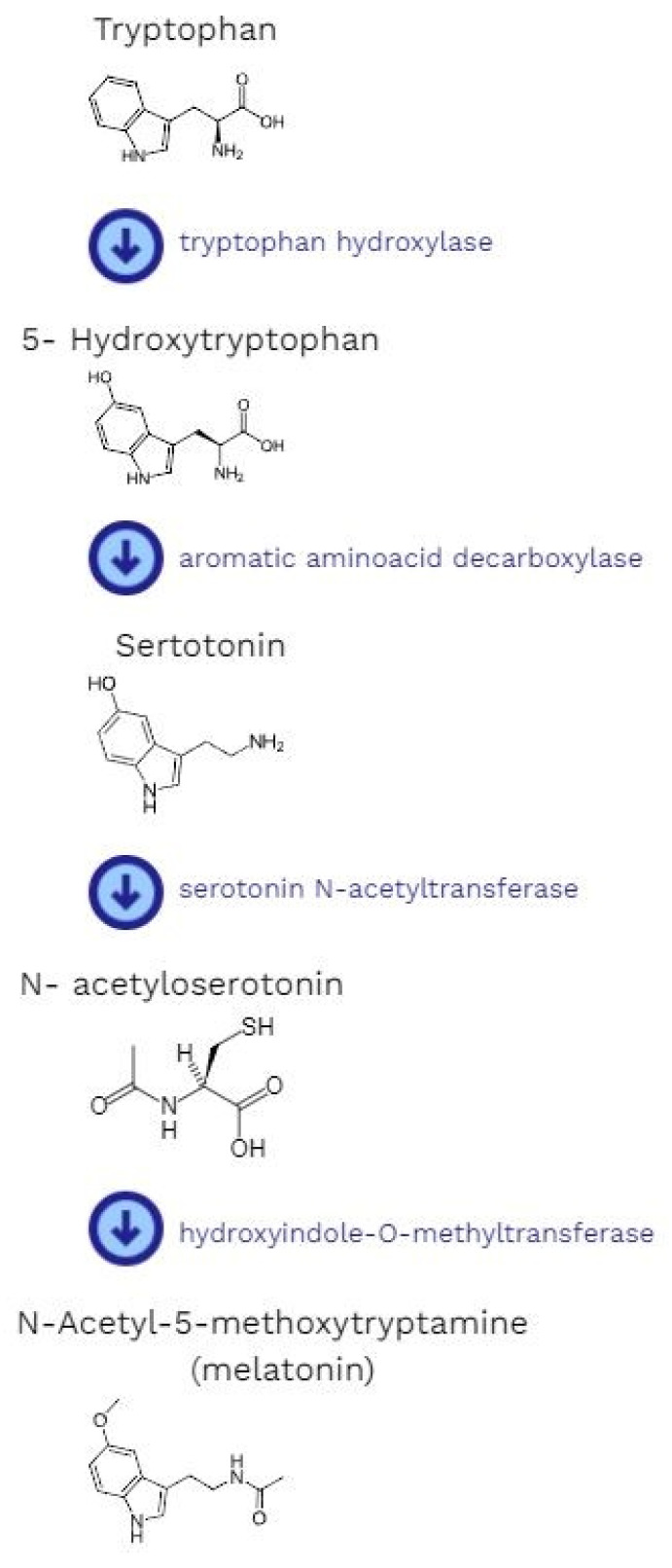
The pathway of melatonin synthesis.

**Figure 2 cells-12-00838-f002:**
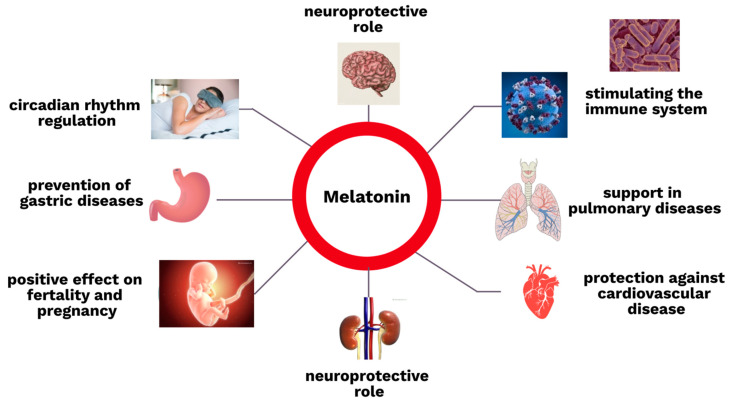
The roles of melatonin in the human body.

**Figure 3 cells-12-00838-f003:**
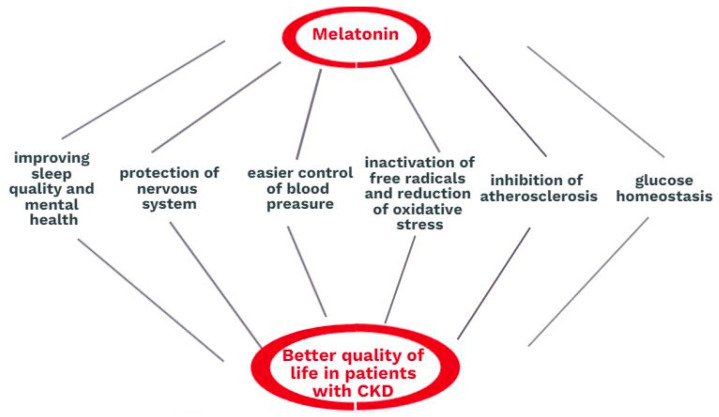
The role of melatonin in kidney diseases.

**Table 1 cells-12-00838-t001:** Morphological alterations in the kidneys, depending on the time of ischemia.

15 Min of Ischemia	25 Min of Ischemia and 120 Min after Blood Reflow	60 Min of Ischemia and 120 Min after Blood Reflow
Insignificant black leak Minimal invertible injury to tubular cells No evolution changes to necrosis	Modest back leak Expansion of cell injury to necrosis in single cells, but convoluted proximal tubules are not involved	Severe black leak Necrosis in convoluted and straight proximal tubular cells, irreparable cell injury, and necrotic cells and sporadic areas of the tubular basement membrane are stripped of epithelium

## Data Availability

No new data were created or analyzed in this study. Data sharing is not applicable to this article.

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
