# Peer review of "Melatonin Treatment in Kidney Diseases"

_cells, 2023, doi:10.3390/cells12060838_

Round 1

Reviewer 1 Report

The manuscript by Markowska et al. encloses interesting and comprehensive insights into melatonin’s role in treatment of kidney diseases. Nevertheless, editorial minor revision and reference-like corrections including the latest ones are strongly encouraged.

Particularly within the introduction section where the author mention about melatonin’s antioxidative, anti-inflammatory, anti-apoptotic and immune processes (Line 28-29).

 Thus, missing significant citations are essential in this place as follows

  1. Bilska B et al., J Pineal Res. 2021 Apr;70(3):e12728
  2. Kleszczyński et al., J Pineal Res 2019;67:e12610
  3. Tan et al., Molecules 2015;20:18886e906.
  4. Reiter et al., J Pineal Res 2020a:e12677
  5. Reiter et al., Cell Mol Life Sci 2020;77:2527–2542
  6. Reiter RJ et al., Cell Mol Life Sci. 2022 Feb 20;79(3):143
  7. Reiter RJ. Prog Neurobiol 1998;56:359e84.
  8. Slominski et al., J Invest Dermatol 2018;138:490–499
  9. Slominski et al., J Pineal Res 2020;68:e12626
  10. Bocheva G et al., Int J Mol Sci. 2022 Jan 22;23(3):1238
  11. Hardeland R. Endocrine 2005;27:119e30.
  12. Hardeland R. Biofactors 2009;35:183e92.

Figure 1. Respective images of the compounds are optically squeezed perpendicularly and it look strange. Please adapt them accordingly.

Figure 2. Apparently, attached images are pasted from the web.

Authors are encouraged either to use the graphical software to generate unified visualizations or there are missing respective sources which are already visible in the submitted version.

       In my humble opinion, it does not look professional and the impact factor of the journal (IF. 7.666) is already respectful.

Author Response

Dear Sir/Madam,

Thank you so much for your opinion about our manuscript. I think it is very pertinent. 

Point 1. I have added citations in introduction, all that you had mentioned and some extra (not only in this section of manuscript). I have stressed it by using red colour. 

Point 2. I have changed the figures. I agree with your opinion that previous graphs looked unprofessional. Please forgive me, I have done it by myself and i am not good at creating graphics. In Figure 2 images are attached from microsoft word -> cliparts. 

I hope that you will be satisfied with the effect of my work.

Yours faithfully, 

Magdalena Markowska. 

Reviewer 2 Report

Overall, the text is very poor in language and requires a reassessment of grammar and syntax.

In some sentences the subject is missing, others are unclear.

There is also a general lack of references that would be useful to complete.

The abstract also, especially in its initial part, does not provide a complete overview and does not focus well on the purpose of the work.

For the Introduction

- the definition of melatonin as a neurohormone/hormone should be reviewed, not all experts in the field agree on this

- there is no reference to refer to, they must be added.

First paragraph

- again, the definition of melatonin needs to be re-evaluated

- the description of the nervous pathway in which it is involved also needs to be rearranged, it is not well delineated and clear references are missing

- moreover, the part about the melatonin synthesis is very trivial, and can be found exactly the same in other articles. Written in this way, it is redundant.

Second paragraph

- the topic of the paper is the interaction between melatonin and kidney function. I find it redundant to consider the role that melatonin plays individually in relation to individual systems, it would perhaps make more sense to reshape the discussion by writing an even shorter, but more homogeneous and linear paragraph.

- the diagram (fig. 2) depicting the different action points of melatonin, on the other hand, is very useful. The paragraph should also be more concise and straightforward.

Third paragraph

- the work is supposed to be a literature review, but to be considered as such, it must present a larger number of references that have to be integrated

- this central section should be the heart of the work, but instead it is somewhat meagre and not analyzed in detail

- it might be useful to better highlight the role of melatonin in restoring physiological morphology (not only in the case of ischaemia).

- I suggest further investigation about the expression of other proteins present at the cellular level in the nephron (also assessing how their expression changes with or without melatonin treatment).

In the final summary portion, more emphasis could also be given to the purpose of the work, with more emphasis on a possible clinical purpose of the work.

Author Response

Dear Sir/ Madam,

Thank you for your opinion about our manuscript. I have tried to improve it according to your indications. 

  1. Text has been edited by native speaker.  He has also noticed some mistakes and it has been corrected so the level of language should be better now.
  2. I have added extra references, especially in introduction but you can find it also in other sections of manuscript. New references have been stressed by red colour as well as additional sentences in the work.
  3. I have modified information about melatonin synthesis to explain it better.  
  4. I have made some corrections in abstract and i have tried the connection of the role of melatonin in individual systems with kidney function. I hope it is well- argued. 
  5. I have made changes in both graphics.
  6. The main section has been extended. I have added descriptions of influence on physiological morphology in the parts when it has been possible to find it.

If there will be something else that you consider I can do to improve this work I am waiting for your suggestions. 

Yours faithfully, 

Magdalena Markowska.

Round 2

Reviewer 2 Report

There has been a substantial improvement in the literature included.

English has also improved, although the text still requires revision overall.

The abstract has been expanded and is clearer.

The first paragraph also better addresses the topic and gives a clearer view of melatonin synthesis.

The whole text also begins to acquire more "structure".

It might also be interesting to better emphasise the role of melatonin in sleep regulation in patients with progressive renal insufficiency. That is, it would be useful to better emphasise the impact that supplementation could have on the patient's quality of life (also citing some examples if there are already studies).

In the final paragraph, it would also be useful to provide a summary schematic outline of the different ways in which melatonin intervenes in regulating the mechanisms underlying different diseases involving the kidney.

Author Response

Dear Sir/ Madam,

I have made additional modifications. The part about the role of melatonin in sleep regulation in patients with progressive renal insufficiency has been expanded - it is highlighted in red colour. In the final paragraph I have added the diagram gathering effects of melatonin. Also English has been checked once again.

Thank you once again for your sugestions.

Yours faithfully,

Magdalena Markowska.

Round 3

Reviewer 2 Report

The requested changes have been made; it is now much more understandable and clear and in this form the text is acceptable for submission.